# A practical application of generative adversarial networks for RNA-seq analysis to predict the molecular progress of Alzheimer's disease

**Jinhee Park**[1,2], **Hyerin Kim**[1], **Jaekwang Kim**[1], **Mookyung Cheon**[1]*

**1** Dementia Research Group, Korea Brain Research Institute (KBRI), Daegu, Korea, **2** School of Electronics Engineering, Kyungpook National University, Daegu, Korea

* mkcheon@kbri.re.kr

**Data Availability Statement:** Python, jupyter notebook codes, and data are available from GitHub (https://github.com/KBRI-Neuroinformatics).

## Abstract

Next-generation sequencing (NGS) technology has become a powerful tool for dissecting the molecular and pathological signatures of a variety of human diseases. However, the limited availability of biological samples from different disease stages is a major hurdle in studying disease progressions and identifying early pathological changes. Deep learning techniques have recently begun to be applied to analyze NGS data and thereby predict the progression of biological processes. In this study, we applied a deep learning technique called generative adversarial networks (GANs) to predict the molecular progress of Alzheimer's disease (AD). We successfully applied GANs to analyze RNA-seq data from a 5xFAD mouse model of AD, which recapitulates major AD features of massive amyloid-β (Aβ) accumulation in the brain. We examined how the generator is featured to have specific-sample generation and biological gene association. Based on the above observations, we suggested virtual disease progress by latent space interpolation to yield the transition curves of various genes with pathological changes from normal to AD state. By performing pathway analysis based on the transition curve patterns, we identified several pathological processes with progressive changes, such as inflammatory systems and synapse functions, which have previously been demonstrated to be involved in the pathogenesis of AD. Interestingly, our analysis indicates that alteration of cholesterol biosynthesis begins at a very early stage of AD, suggesting that it is the first effect to mediate the cholesterol metabolism of AD downstream of Aβ accumulation. Here, we suggest that GANs are a useful tool to study disease progression, leading to the identification of early pathological signatures.

## Author summary

We applied a deep learning technique called generative adversarial networks (GANs) to bulk RNA-seq data, where the number of samples is limited but expression profiles are much more reliable than those in single cell method. Like continuous image conversions of human faces commonly used in the recent AI revolution, we introduced virtual

**Funding:** This research was supported by KBRI basic research program through Korea Brain Research Institute funded by Ministry of Science and ICT (20-BR-02-09 (JK) and 20-BR-02-10 (JP, HK, MC)) and by the Korea Health Technology R&D Project through the Korea Health Industry Development Institute (KHIDI), funded by the Ministry of Health and Welfare, South Korea (grant number: H I14C1135) (MC). The funders had no role in study design, data collection and analysis, decision to publish, or preparation of the manuscript.

**Competing interests:** The authors have declared that no competing interests exist.

Alzheimer's disease progression described by gene expression levels. Our gene expression analysis based on GANs is proposed to capture pathological pathway cascades and sequential orders of gene regulation. Through this convergence study of bioinformatics and AI, we discovered that amyloid-beta production is thought to trigger the cholesterol biosynthesis.

## Introduction

Data science using deep learning, which is a central approach driving the revolution in artificial intelligence (AI) technology in recent years, has been extensively applied to genomics [1–9]. Omics-based studies inspired by the advancement of next-generation sequencing (NGS) technology have become easily accessible and standardized, and an increasing amounts of omics raw data are being publicly released. Taking full advantage of this accumulated NGS data, we expect convergence studies involving deep learning and bioinformatics to become highly influential and provide novel breakthroughs in analyzing data and capturing insights.

In 2014, a new deep learning method called a generative adversarial network (GAN), which can generate and manipulate fake data, received considerable attention in image generation studies [10,11]. Over the years, many technological advances have been made to create fake data that are increasingly difficult to distinguish from real images [12]. Notable developments have also been made to transfer style from one picture to different images [13]. Brief and general explanations for GANs are given in Materials and Methods.

GANs have the potential to perform manipulations such as smooth transitions between images or to implement certain features through vector arithmetic in the latent space, which has been a key for application of GANs in image analysis [11,13]. Amazingly, Ghahramani *et al.* achieved this idea in a single-cell RNA-seq (scRNA-seq) study of epidermal cells [9]. They could perform gene expression simulations to describe the cell differentiation process through latent space interpolation. This application inspired us to apply it to analyze disease progress by capturing pathological pathway cascades, which may be useful in identifying early biomarkers for AD causal factors.

However, numerous genes with noisy expression levels in scRNA-seq data would increase the difficulty of generating high-quality fake data, causing inaccurate initial and final states in the latent space interpolation. Therefore, simulating gene expression changes to describe cell differentiation or biological pathway cascades seems dependent on having sufficient qualified data.

But what if we were to apply the GANs to perform continuous conversions of gene expression levels to bulk mRNA-seq data with fewer noisy attributes? In this case, datasets with many biological contextual samples for practical use are lacking. This data shortage substantially hinders deep learning applications in transcriptomics except for single-cell studies. Nonetheless, it is worthwhile to develop a transient methodology to circumvent this obstacle under the expectation that we will have access to high-quality data in the future.

To apply this technique to bulk RNA-seq, we need to carefully reinspect tricky issues such as the number of samples, the number of genes used for learning, and data normalization. Due to the labor-intensive process involved in sample preparation and the sequencing cost for typical bulk RNA-seq data, the number of samples per condition in contexts such as age, phenotype and treatment will inevitably be small compared to the scRNA-seq data. General transcriptome studies rarely exceed dozens of samples per condition for animal data under well-controlled conditions or hundreds of samples for human tissue data with a variety of

intrinsic heterogeneity [14]. In other words, given a relatively small number of samples, it is impossible to apply GANs to bulk RNA-seq profiles. This situation motivated us to devise a data augmentation method to artificially increase the number of samples in the training data. However, the lack of statistical power due to the small number of available sample data limits the ability to generate reliable gene expression profiles. Hence, it is not easy to generate fake profiles that include more than tens of thousands of genes expressed in reference annotations. The larger the number of genes we use in GANs is, the lower the reliability of the obtained fake gene expression profiles is. Thus, we should not include all the expressed genes; instead, we should select only the differentially expressed genes (DEGs) that passed a significant test. This approach leaves thousands or fewer genes for use in machine learning. Most scRNA-seq studies use TPM (transcripts per million reads) units considering normalization of library size, but in bulk mRNA-seq studies, we can use better normalization by DESeq2, which considers library size and composition together [15]. Additionally, after normalizing the gene expression counts, we need to rescale them again to fit the input scale for deep learning applications.

In this paper, we applied the GANs not to scRNA-seq but to bulk RNA-seq data with fewer variations in gene expression levels and a smaller number of genes. We targeted the Alzheimer's disease (AD) model of mouse data, which are publicly available and homogeneous under a well-controlled context, to perform a gene expression simulation from wild type (WT) to AD states with the purpose of discovering pathological pathway cascades or causality. Pathological pathway analysis based on the evolving curve patterns of gene expression is proposed to capture orders of gene regulation and coexpression to interpret disease progression. As an example, we dissect cholesterol biosynthesis as an early event due to enhanced amyloid-beta production.

## Results

### Preprocessing for applying the GANs to bulk mRNA data

Fig 1 shows an overview of the application of the GANs to bulk RNA-seq data. As a qualified dataset for AD mechanism study, we selected the cortex data of WT and AD (5xFAD) phenotype mice, which are known to form amyloid-beta (Aβ) plagues, at three different ages (2M, 4M, and 7M) from a GSE104775 dataset released to the NCBI GEO website [16]. Using the commonly used RNA-seq pipeline (Trimmomatic-HISAT2-HTSeq-DESeq2) [15,17–19], we were able to obtain significant DEGs (1,208 genes (7M WT vs 7M AD), 1,193 genes (4M WT vs 4M AD) and 3 genes (2M WT vs 2M AD)) filtered by an adjusted p-value < 0.05 (Benjamini-Hochberg procedure) and the DESeq2 scaled counts (the mean scaled counts >100) (S1 Table). Only 1,208 DEGs were selected for training the GANs because we want to focus on the pathological pathway analysis. We normalized the gene expression counts and transformed them into regularized logarithmic data (RLD) using DESeq2. Based on 36 real sample data, we devised a data augmentation procedure for RLD and obtained a total of 846 samples (see Materials and Methods). The DESeq2 normalized data (RLD) of 1,208 genes was rescaled so that 95% of the values were within (0, 1); the unit of the rescaled RLD is appropriate for deep learning (see Materials and Methods).

### The training process and generation of fake gene expression profiles

We employed the WGAN+GP algorithm [20,21] used in Ghahramani's scRNA-seq analysis [9]. The network architecture was reshaped, and the hyperparameters were adjusted for the current number of targeted genes (see Materials and Methods). Of the 846 augmented samples, 762 were randomly selected as a training set, and the remaining 84 samples were used as a test set. The learning process was conducted for up to 200k epochs, which is 10 times more

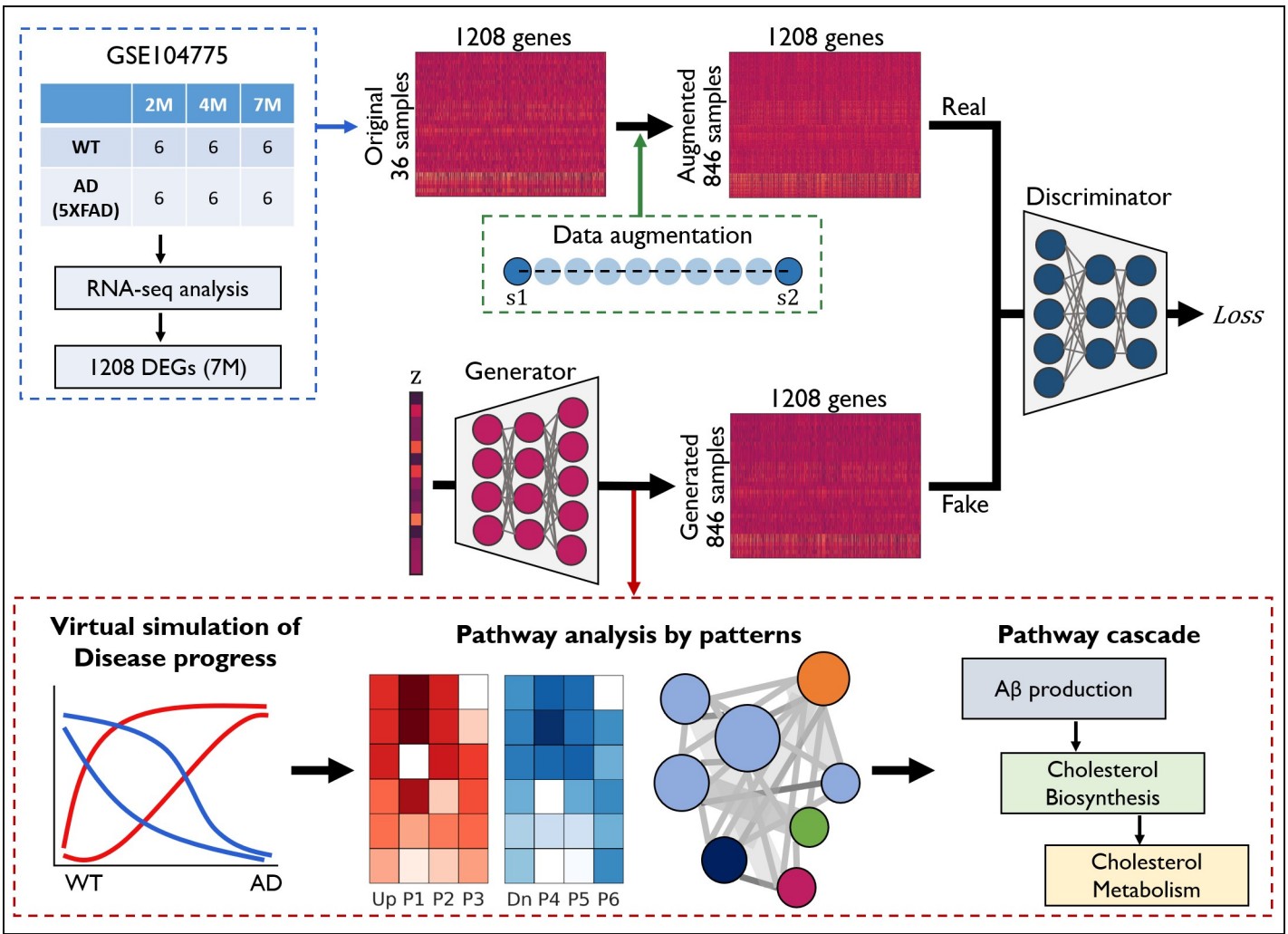

**Fig 1. Overview of the application of the GANs to bulk RNA-seq data.** RNA-seq analysis for the GSE104775 raw data with 36 WT and AD samples (n = 6/group) was performed, yielding 1,208 DEGs between 7M WT and 7M AD. The normalized expression profile for the 36 samples was subjected to a data augmentation procedure, creating 846 augmented samples. The generator network produces fake gene expression data with random variables in a latent space(z). The discriminator network distinguishes between the augmented real and fake data to yield a loss function applied to the training weight parameters of both networks. The transition curves for the 1,208 gene expressions change between WT and AD, showing a virtual simulation of disease progress. Then, these are evaluated by latent space interpolation with the generated fake data. The transition curves were classified into six patterns (P1 to P6) to perform pathway analysis with gene lists of pattern subsets. We identified the order of up- or downregulated pathways that predict the pathway cascades.

than the training time of the scRNA-seq work, and used the Adam optimizer. We observed that training convergence was reached after 25k epochs (loss values from S1A Fig) and similarity between the generated fake data and the real data has reached a maximum after 75k epochs and has been saturated (S1B Fig).

A comparison of the distribution plots of the normalized gene expression values (rescaled RLD) for the same number of fake samples as the 846 augmented real samples at 100k epoch, where the similarity of the generated data is after the maximum, shows that two distributions are almost identical (Fig 2A), which indicates that our new normalization working very well. We also plotted the Pearson correlation distributions within the augmented real samples and between the real and generated data at the 100k epoch (Fig 2B). The distribution patterns are

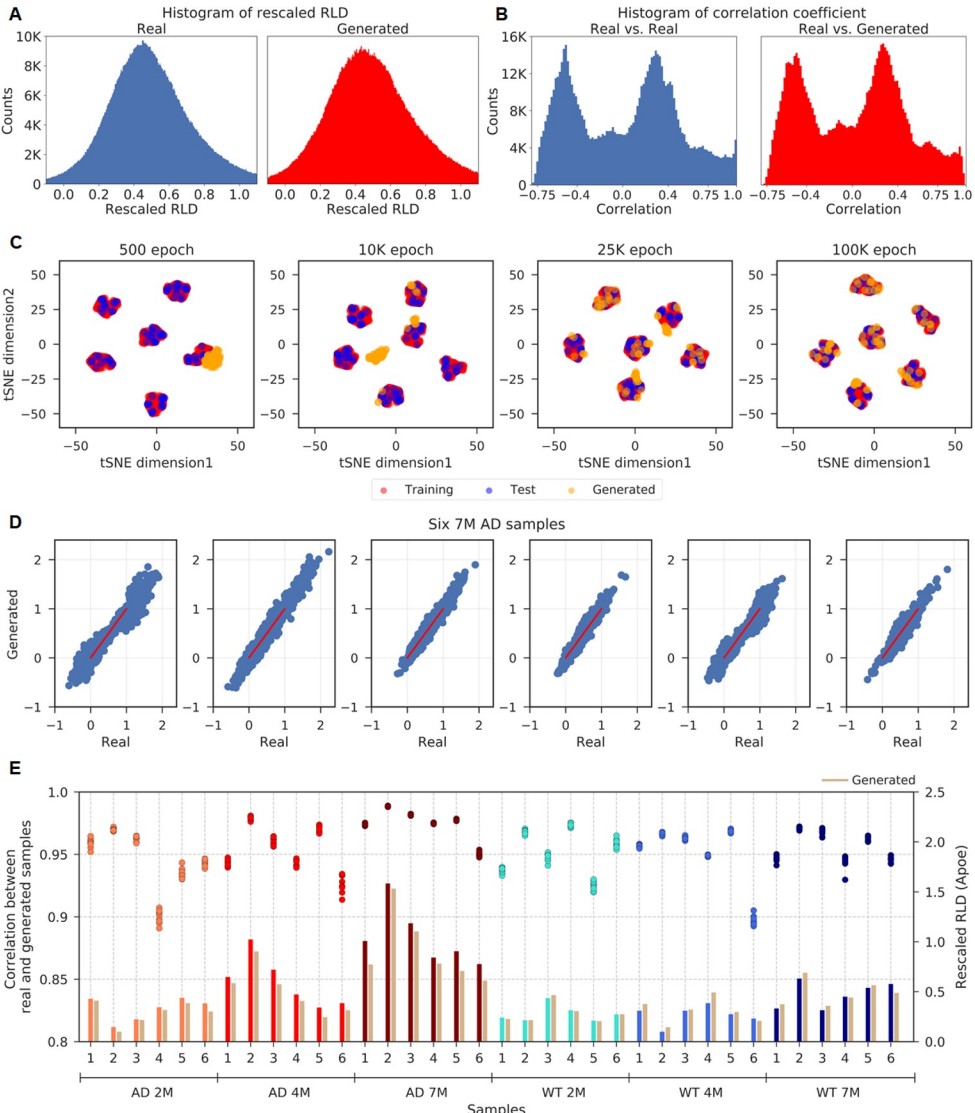

**Fig 2. Generation of fake gene profiles.** (A) The distribution plots of all rescaled RLD values for the 846 augmented real samples (blue) and the 846 generated samples (red). 95% of the real and 93% of the generated values lie within [0, 1]. (B) Correlation coefficient distributions for all pairs within the 846 real data (blue) and for pairs between the 846 real and 846 fake data (red) at the 100k epoch. The two broad peaks represent the correlation coefficients within the same group and between different groups. (C) tSNE plots at four epochs with different colored dots representing the 762 training (red) samples, 84 test (blue) samples and 84 generated samples (orange). Remarkably, we observed well-separated clusters per group in the tSNE plots due to the process of pairwise data augmentation performed within a group. (D) The scatterplots of the rescaled RLD values of the 1,208 genes for six 7M AD samples vs the corresponding resembled fakes. (E) The rescaled RLD values of the Apoe gene for 36 samples and their resembled fakes (generated; mauve color), which were generated at the 100k epoch. The scattered points of the correlation coefficients stand for different evaluations over ten repeated generations of 10,000 fake data. Low variations of correlation coefficients indicate that the fake data averaged in the latent space seem to be robust and have similar values under the given weight parameters of the generator network.

highly similar except for the self-correlation values, which show a sharp peak at 1. T-distributed Stochastic Neighbor Embedding (t-SNE) [22] plots at four epochs reflect the progressive learning process (Fig 2C). The generated data (orange) are together at training onset (500 epochs). After training convergence (from the 25k to the 100k epochs), the data are well

clustered with the real data corresponding to six condition groups. The comparison plots of the rescaled RLD distributions and tSNE plots strongly support the idea that the GANs can successfully generate fake gene expression profiles for the selected genes (DEGs) of WT and AD model mice.

## The similarity of the generated data to real samples

To obtain fake data that is as similar as possible to the real data, we generated 10,000 fake data at a given epoch and compared them with the 36 original real samples. We selected the top 10 generated data with the highest correlations for each of the 36 samples and obtained a fake per sample by averaging the ten latent space variables, which produced 36 fake expression data corresponding to the real samples. This averaging process over similar fake data is the same process used in GANs applications for face images, which require to get stable manifolds to show the semantic changes [11]. We termed the averaged fake sample corresponding to a specific real sample, which was obtained from 10,000 generated data, as a "resembled fake". The scatter plots of 1,208 genes are presented to show the high Pearson correlations (r = 0.95~0.98) between the original real data and the resembled fakes (Fig 2D). Specifically, we examine the resembled fake values of individual genes. The rescaled RLD values of the Apoe gene, which plays a crucial role in AD pathogenesis [23], are shown to have high accordance between the 36 real and the resembled fake data (Fig 2E). The additional tSNE plots for the original(36), the augmented(810) and the resembled fakes(36) corresponding to the original samples show that visualization of clustering is enhanced by pairwise data augmentation within a group and the resembled fakes are well overlapped to the original real samples (S2 Fig). As a result, if we generate 10,000 fake data at a given epoch time, the correlation coefficients of the resembled fakes will be between 0.89 and 0.98 for the 36 original real and between 0.83 and 0.98 for 84 test data.

## Sample-specific or gene-specific features for collective gene association in the generator

We investigated the output data trends and network parameters of the GAN generator to identify how the network extracts biological features. The resembled fakes appear to be highly similar to the corresponding specific samples (Fig 2E). However, they continue to change and fluctuate over epochs even after reaching the optimized basin (S3 Fig). We investigated gene associations based on the temporal deviations of the resembled fakes (S4 Fig Top panels) and suggest that the generator becomes trained on the collective behaviors of gene expressions, which is supposed to contain sample-specific features rather than gene-specific features. We also confirmed the gene-specific collective behaviors in the gene association network (S4 Fig Bottom), which are expected to be reflected by the weight parameters of the last layer in the generator. We expect that these network features can produce biological semantic changes when we apply the latent space interpolation to descriptions of disease progress.

## Latent space interpolation describing disease progress from WT to AD

Here we can apply the latent interpolation in two directions, which are transitions from 2M to 7M in the direction of aging or from WT to AD in the direction of disease progression. In this study, we focus on the latter transition, which is not accessible from traditional bioinformatics analyses. In the AD mouse model (5xFAD), specific transgened human genes (PSEN1, APP) induce a disease state through increased Aβ production. In other words, these mice tend to develop the disease following specific pathways such as the familial AD pathologies. Here, we need to approach the topic from a new perspective. If we have the final AD gene expression

state—e.g., activated immune cells, neurite deficiencies, etc.—by following the methods of image interpolation [11], the latent vector arithmetic of the GANs would simulate a virtual disease progression from the WT to the AD state. That is why we focus on the transition from WT to AD. Although the AD state is caused by the Aβ overproducing mouse model, the coexpression network of the trained GANs might search for semantic changes that capture the disease progression. Our hope is that this GAN application will describe a virtual disease progression and provide novel perspectives regarding pathological cascades that are hard to find using the typical bioinformatics approach of controlling the aging process [16].

The smooth transitions between the two states are evaluated through the vector arithmetic equation in latent space. By averaging the procedure over the same states and epochs, we obtained the transition curves for 1,208 genes from WT to AD for 2M, 4M and 7M (see Materials and Methods). For examples, we selected only 17 genes, which are known to be highly related or have similar names, for plotting their transition curves on one axis from 7M WT to 7M AD (Fig 3A and 3B) and three axes (2M WT to 2M AD, 4M WT to 4M AD, and 7M WT to 7M AD) (S5 Fig). We present the transition curves in RLD scale for 17 genes with the expression values for real 36 samples (three colored dots).

The curve patterns in each image in Fig 3A and 3B are highly correlated, which indicates that coexpressed genes of causal biological processes follow similar patterns on the path toward virtual disease progress. Some curves (Mdh1, Mdh2) show gradual changes, but some others (Apoe, Abca1) show significantly delayed increases. These observations suggest that genes can be classified by patterns, which will aid in identifying temporal gene regulation or causal gene association to clarify pathological pathways.

We classified the transition curves of the 1,208 genes into six patterns (Fig 3C). Each transition curve can be subjected to one-to-many correspondence by measuring the correlations with six predefined patterns (the red lines in Fig 3C). The patterns are categorized according to whether the curve changes involve mainly early-stage changes (patterns 1 and 4), gradual changes (patterns 2 and 5), or late-stage changes (patterns 3 and 6). The first three patterns (patterns 1–3) are transition curves for the upregulated genes, and the last three patterns (patterns 4–6) are for downregulated genes. The Venn diagrams for the number of genes belonging to each pattern show that some genes belong to two patterns, indicating that the genes in the intersection of the sets have intermediate patterns between the two (Fig 3D).

## Pathological pathways

We performed gene ontology (GO) and pathway analysis with the 1,208 genes (7M DEGs) using WebGestalt, an integrated analysis toolkit website for biological contexts [24]. With gene lists classified into upregulated (Up), downregulated (Down), and six subsets (P1, P2, P3, P4, P5, and P6) according to their patterns (Fig 3D), we performed pathway analysis based on two pathways (KEGG & WikiPathway) and the GO biological process (GOBP) provided by WebGestalt. We selected 50 pathways that are thought to be relevant biological processes in AD for upregulated (Fig 4) and downregulated genes (Fig 5). The false discovery rate (FDR) and enrichment ratio (ER) values for the selected pathways are plotted in Fig 4 (Up, P1, P2, and P3) and Fig 5 (Down, P4, P5, and P6). We also plotted the heatmaps for the average transition curves of pathways with genes belonging to the UP or DOWN lists (Figs 4B and 5B). The full list of the pathways that reached significance (Benjamini-Hochberg FDR <0.05) are shown in S2 Table.

The dominant pathways with upregulated genes are related to inflammatory systems: TYROBP causal network (WP3625, FDR = 0), microglia pathogen phagocytosis pathway (WP3626, FDR = 7.7E-14), chemokine signaling pathway (mmu04062, FDR = 3.6E-3), complement and

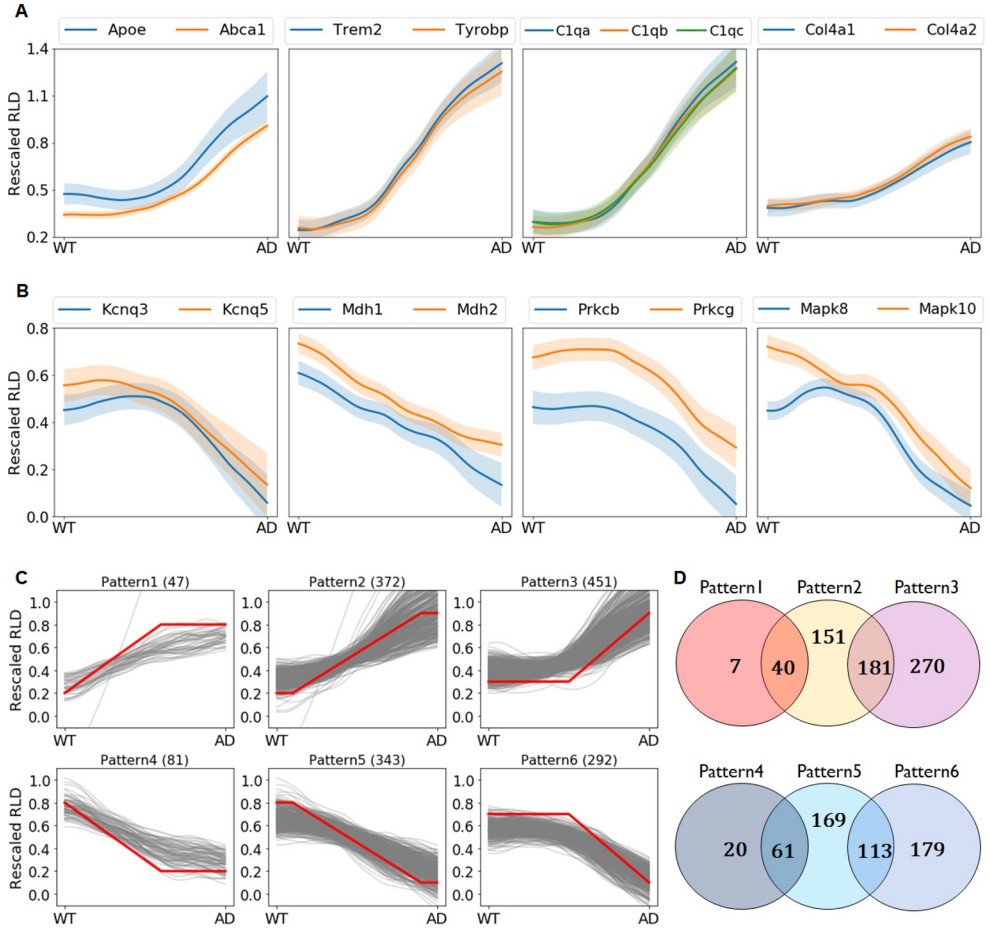

**Fig 3. Transition curves of gene expression levels.** (A-B) Transition curves of selected 17 genes from 7M WT to 7M AD: some up- and downregulated genes, which are known to be highly related or have similar names, were selected from pathways such as cholesterol metabolism (Apoe, Abca1), microglia pathogen phagocytosis pathway (Trem2, Tyrobp), complement and coagulation cascades (C1qa, C1qb, C1qc), focal adhesion (Col4a1, Col4a2), cholinergic synapse (Kcnq3, Kcnq5, Prkcb, Prkcg), TCA cycle (Mdh1, Mdh2), and dopaminergic synapse (Mapk8, Mapk10); (C) Each curve belongs to a pattern when r (the correlation coefficient with the predefined red colored curve patterns which we proposed) is higher than 0.95 or when the maximum r is above 0.90; (D) Venn diagrams for the number of transition curves belonging to each pattern. The total number of curves in the six patterns is 1,191 (649 upregulated, 542 downregulated). Among the 1,208 DEGs, seventeen genes could not be classified into the six patterns.

coagulation cascades (mmu04610, FDR = 7.0E-4), NF-kappa B signaling pathway (mmu04064, FDR = 9.1E-3), cytokine-cytokine receptor interaction (mmu04060, FDR = 0.039), etc. in 25 selected UP pathways. We observed many inflammatory GOBP pathways with named leukocytes (S2 Table) not shown in the 25 UP pathways. The next noticeable pathways are associated with integrin-associated pathways: the integrin-mediated signaling pathway (GO:0007229, FDR = 3.6E-16), cell adhesion mediated by integrin (GO:0033627, FDR = 7.5E-8), and focal adhesion (WP85, 1.0E-3), which receive attention in AD pathogenesis [25]. The dominant pathways with downregulated genes are related to vesicles and neurotransmitters: synaptic vesicle cycle (mmu04721, FDR = 7.3E-5; GO:0099504, FDR = 2.1E-8), neurotransmitter transport (GO:0006836, FDR = 2.4E-8), exocytosis (GO:0006887, FDR = 7.4E-8), and vesicle-mediated transport in synapses (GO:0099003, FDR = 4.1E-10). The next noticeable pathways with downregulated genes are associated with specific synapses: GABAergic synapse (mmu04727, FDR = 1.3E-3), dopaminergic synapse (mmu04728, FDR = 1.8E-3),

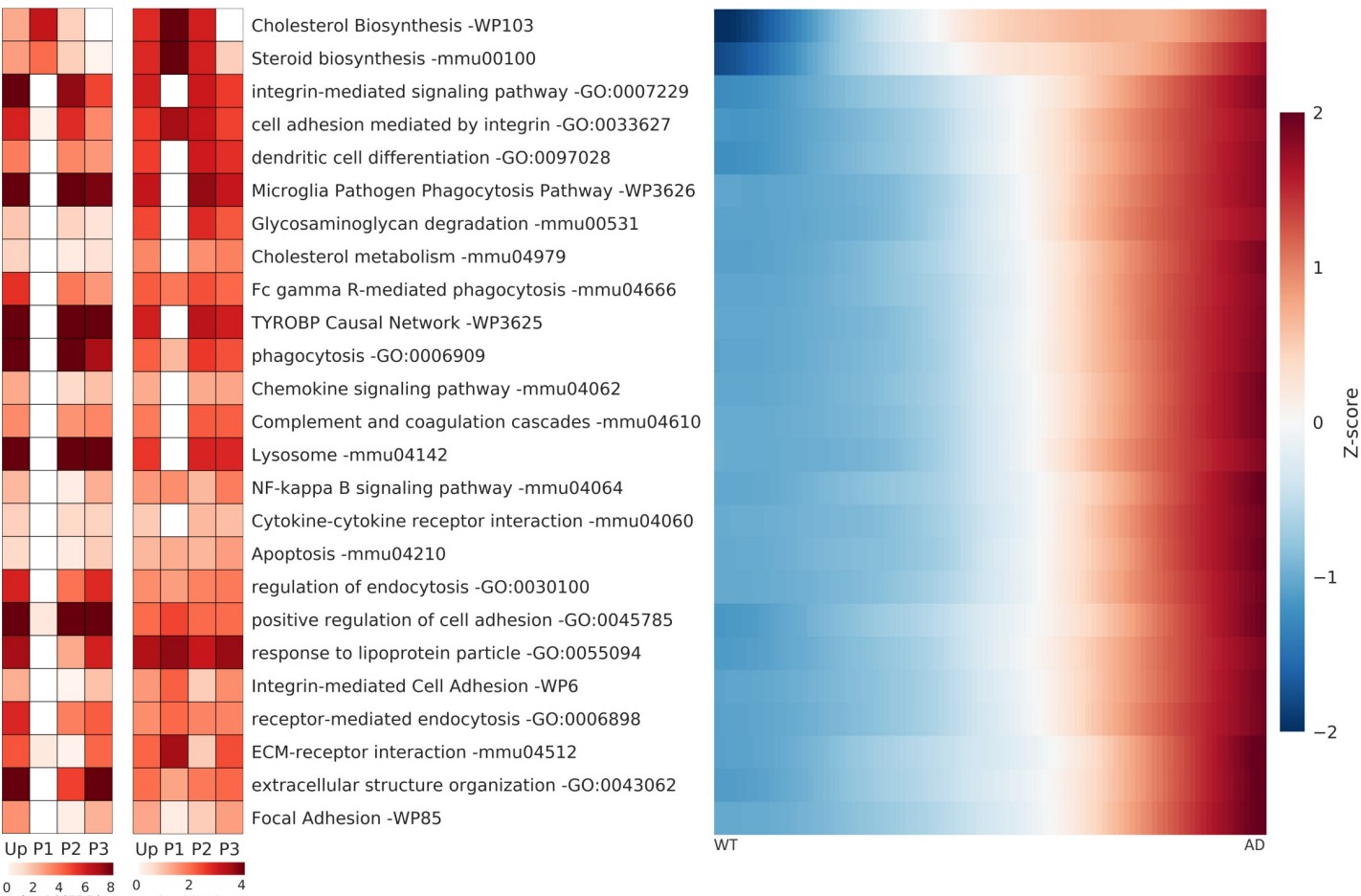

**Fig 4. The false discovery rate and enrichment ratio value for upregulated pathways.** The false discovery rate (FDR) and enrichment ratio (ER) values for the selected pathways, which were estimated by the gene list of each subset (Up, P1, P2, P3). Each pathway's heatmap was estimated by averaging the transition curves of the genes belonging to the Up list. Some pathways exist in which the ER values are very large but whose FDR values are not significant. This is the reason why the number of genes in subset lists such as P1 and the annotated genes in some pathways (such as cell adhesion mediated by integrin) are small; consequently a few overlapping genes seem to enhance the ER values. Conversely, the pathways with many annotated genes tend to have very small FDR values even at low enrichment ratios, such as phagocytosis. Hence, we present both the FDR and ER simultaneously.

glutamatergic synapse (mmu04724, FDR = 1.8E-3), and cholinergic synapse (mmu04725, FDR = 1.0E-2). All of these pathways are widely known to be associated with AD pathogenesis [25,26].

A pathway analysis with gene lists of subsets by six patterns, which are expected to show the disease progress, revealed which pattern is the most dominant for each pathway. Cholesterol biosynthesis and steroid biosynthesis are the most dominant in the P1 pattern, suggesting that alterations in cholesterol biosynthesis are likely to occur in the very early stage of AD. Most upregulated pathways not only dominate in one pattern but share genes in two patterns, P2 and P3. For example, TYROBP Causal Network (FDR = 2.2E-12 in the P2 list, 6.7E-10 in the P3 list) has roughly similar FDR values in both patterns. The transition curves of each gene in these pathways are presented in S6 Fig to show the heterogeneity of the genes that follow patterns 2 and 3. However, we observed slight differences among the pathways regarding which genes belong to patterns 2 or 3. The microglia pathogen phagocytosis pathway (3.8E-14 in P2, 2.2E-8 in P3) has lower FDR values in P2, supporting only gradual increases in the transition

curves of each gene (S6 Fig). The ECM-receptor interaction (0.73 in P2, 8.2E-5 in P3) has lower FDR values in P3, indicating later increases in the transition curves (Fig 4 and S6 Fig).

For downregulated genes, the TCA cycle (FDR = 0.45 in P5) is dominant in only P5, and long-term potentiation (0.030 in P6) is dominant in only P6; these show somewhat homogeneous patterns (S6 Fig). Although the two pathways have only five genes in DEGs, their homogeneous patterns indicate that the TCA cycle is a gradual decrease and long-term potentiation is a late decrease. Regulation of the neurotransmitter levels (9.8E-7 in P5, 0.098 in P6), neurotransmitter transport (9.8E-7 in P5, 0.025 in P6), exocytosis (1.9E-6 in P5, 0.044 in P6), synaptic vesicle cycle (9.8E-7 in P5, 0.11 in P6), and vesicle-mediated transport in synapse (2.0E-8 in P5, 2.9E-4 in P6) have significantly lower FDR values P5 (Fig 5). These pathways indicate that the genes associated with exocytosis of neurotransmitters by vesicles are downregulated gradually during the disease progression. Dopaminergic synapse (0.35 in P5, 0.012 in P6), synapse organization (0.34 in P5, 0.021 in P6), cognition (0.16 in P5, 1.0E-3 in P6), glutamatergic synapse (0.12 in P5, 3.0E-3 in P6), locomotory behavior (0.068 in P5, 6.83E-4), cholinergic synapse (0.73 in P5, 0.020 in P6), dendrite development (0.18 in P5, 5.8E-3 in P6), and the Wnt signaling pathway (0.85 in P5, 2.1E-3 in P6) have lower FDR values in P6 (Fig 5). Collectively, these

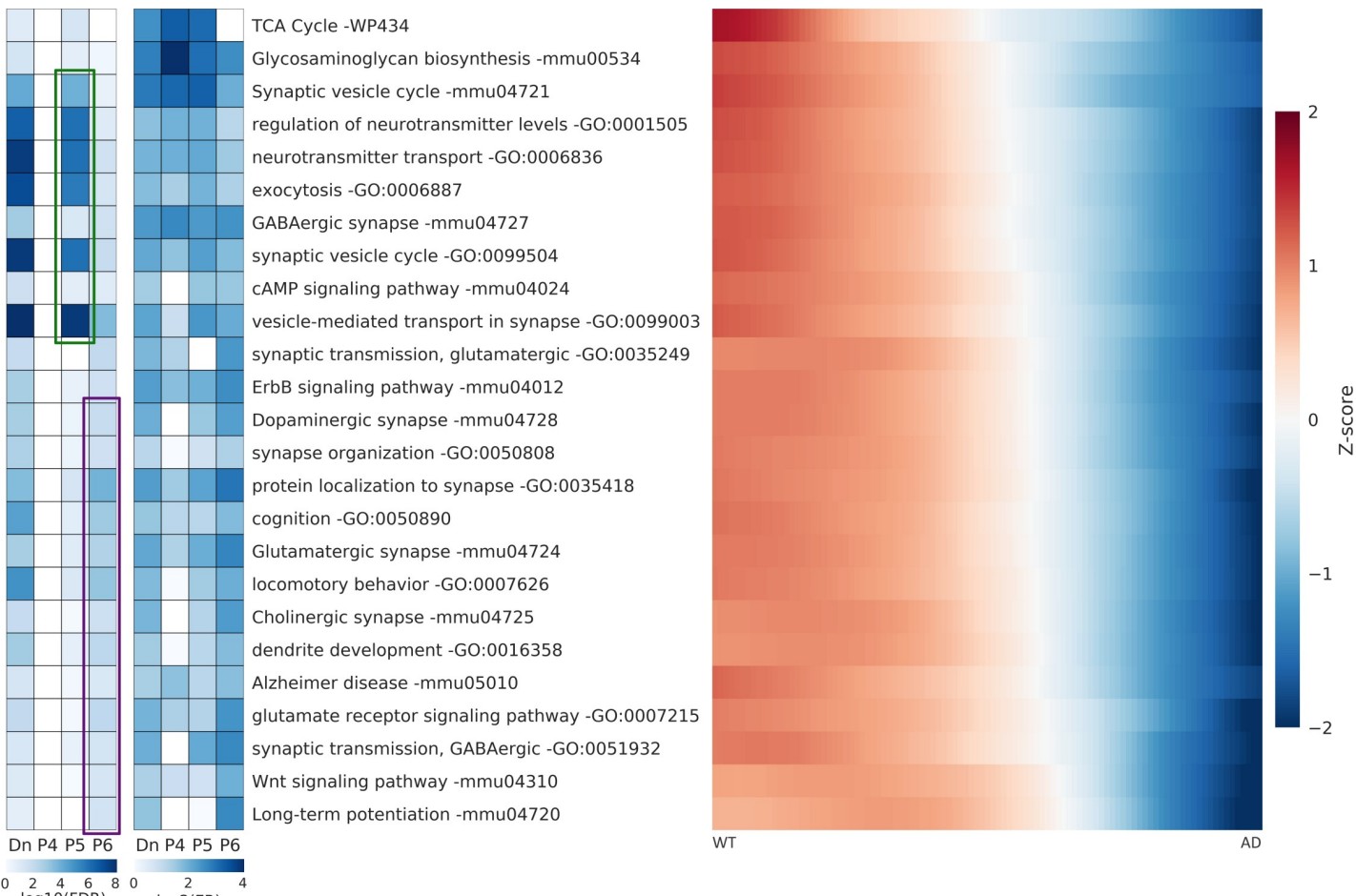

**Fig 5. The false discovery rate and enrichment ratio value for downregulated pathways.** The false discovery rate (FDR) and enrichment ratio (ER) values for the selected pathways estimated by the gene list of each subset (Down, P4, P5, P6). Each pathway heatmap was estimated by averaging the transition curves of the genes belonging to the Down list. Genes were significantly downregulated for pathways involving the exocytosis of neurotransmitters with lower P5 FDR (the green box) and of specific synaptic functions with lower P6 FDR (the purple box).

results indicate that the genes related to specific synaptic functions are downregulated at the late stage of disease progression. It should be noted that the AD model mice are only seven months old; thus, they may not be representative of the severe late state of human AD.

## Cholesterol biosynthesis & cholesterol metabolism

The most surprising finding in the pathway analysis through patterns is that the transition curves of genes of the cholesterol biosynthesis pathway follow pattern 1, suggesting an early change in AD disease progress (Fig 6A and 6B). Of the 15 genes involved in cholesterol biosynthesis (WP103, S7A Fig), five of six upregulated DEGs follow pattern 1, resulting in an extremely higher ER value (ER = 6.7 in UP, 72.4 in P1) and a lower FDR value (FDR = 3.89E-3 in UP, 6.26E-7 in P1). Seven upregulated DEGs belonging to the cholesterol metabolism (mmu04979, FDR = 0.047 in UP) in KEGG pathways showed different increasing patterns

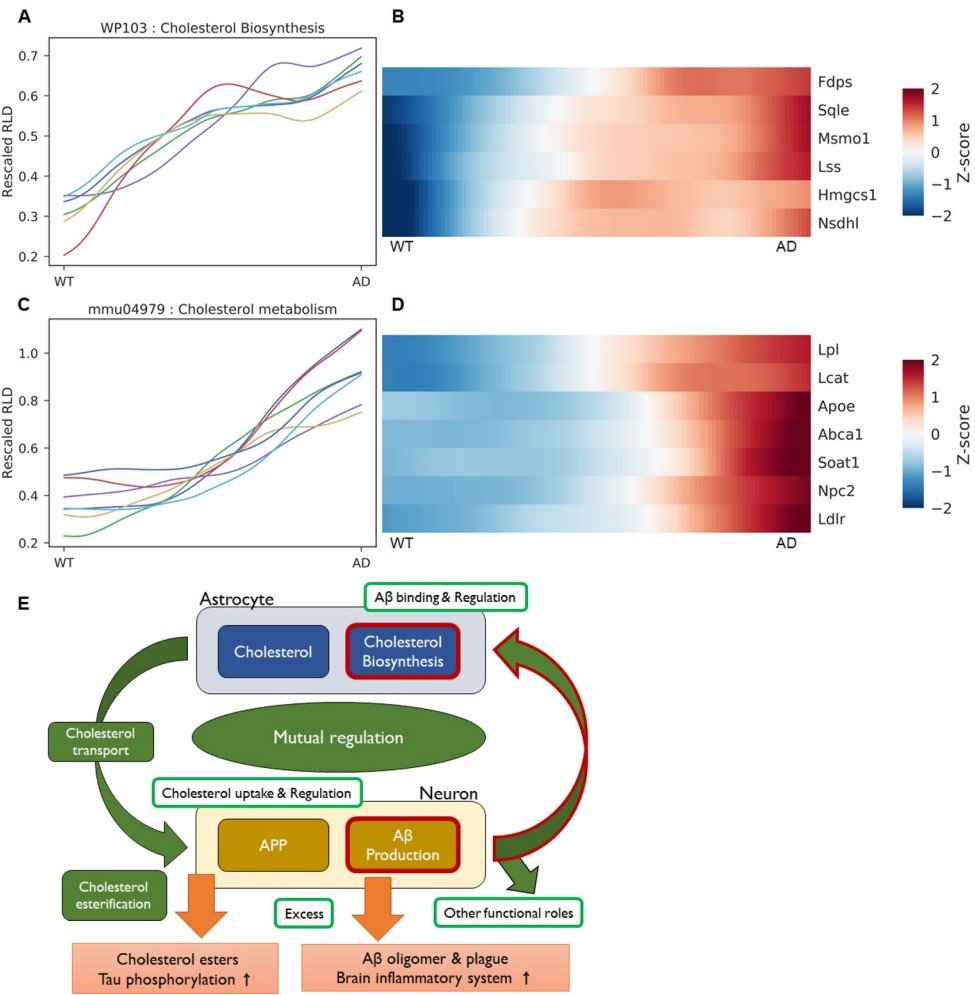

**Fig 6. The transition curves and heatmap of genes related to cholesterol biosynthesis and cholesterol metabolism.**
(A-B) The transition curves of genes in the cholesterol biosynthesis pathway show very early increases and saturation. (C-D) Cholesterol metabolism, which is a KEGG pathway associated with cholesterol, shows a lower FDR value in P3 and a late increase of transition curves for genes such as Apoe, Abca1, and Ldlr. (E) Schematic diagram of a suggested hypothesis for mutual regulation between Aβ production and cholesterol biosynthesis. Our finding that Aβ regulates cholesterol biosynthesis is denoted by the red outlines. Other parts were constructed based on the referenced literature.

following patterns 2 or 3 (Fig 6C and 6D). This suggests that changes in cholesterol biosynthesis occur earlier than do those of cholesterol metabolism.

Interestingly, the transition curves and the heatmaps of Apoe, Abca1, Ldlr, and Soat1 (also known as Acat1) genes show similar late increase patterns. Apoe plays an essential role in cholesterol transport and cholesterol homeostasis. Abca1 and Ldlr mediate cholesterol efflux and influx, respectively. Soat1 (Acat1) and Lcat are genes of the enzymes that esterify cholesterols, and Soat1 has been suggested as a therapeutic target for AD [27]. Therefore, it is apparent that enhancement of cholesterol transport and esterification occur after the enhancement of cholesterol biosynthesis. This early increase is expected to be inherent in the original gene expression profiles, which were confirmed by evaluating the fold changes (S7B Fig) of six DEGs. Noticeably, changes in the genes even after two months were observed to show weak upregulation: the p-values of five genes were less than 0.1, although their adjusted p-values showed no significance (S1 Table). Because the AD (5xFAD) model mice have enhanced production of Aβ by mutated human APP and PSEN1 genes, we can hypothesize that Aβ production would strongly and directly correlate with cholesterol biosynthesis.

Our finding that enhanced Aβ production has a strong correlation with cholesterol biosynthesis was verified by the other two RNA-seq data. We downloaded and analyzed the RNA-seq data from another AD model mouse with tau pathology (PS19-P301S) (GSE90693)[28] and human fusiform gyrus (GSE125583)[14]. Interestingly, the WikiPathway analysis using DEGs from the cortices of six-month-old mice showed downregulation of cholesterol biosynthesis (FDR = 1.56E-5) with seven related DEGs (Dhcr7, Fdft1, Hmgcr, Idi1, Mvd, Nsdhl, and Sqle) and is also consistent with the observation of downregulation of App gene expression (log2FC = -0.168) for the tau mice. It means cholesterol biosynthesis is not positively correlated with tau pathology, supporting our finding that Aβ over-production in AD is a causal factor for enhanced cholesterol biosynthesis. RNA-seq analysis from human fusiform gyrus data from AD (n = 219) and controls (n = 70), which can fluctuate widely depending on the individual, showed no significance in cholesterol biosynthesis between AD and control tissues. However, we observed a positive linear relationship in the expression levels between the human APP and four in the six genes, regardless of AD status (S8 Fig). These results with the additional other RNA-seq analyses support the hypothesis that Aβ production is strongly correlated with cholesterol biosynthesis.

## Discussion

Why is it important for the enhancement of cholesterol biosynthesis to have a high correlation with Aβ production and to occur at a very early stage in disease progress, and how does this finding help us understand AD pathogenesis? Considering cholesterol's roles and metabolism related to AD [29–31], an unusual early enhancement of cholesterol biosynthesis, unlike other pathways, including cholesterol metabolism, suggests that cholesterol biosynthesis can be promoted by direct interactions between cell membranes and Aβ. These interactions are thought to occur in astrocytes because cholesterol biosynthesis is not efficient in neurons—their myelination process is almost complete in mice at 2 months of age [32,33]. It should be noted that some early studies showed inhibition of cholesterol synthesis by Aβ in neuronal or embryonic fibroblast cells [34,35]. We also checked the eight cell-type transcriptome of mouse cortex (GSE52564)[36], showing high expressions of five genes in astrocytes (S9 Fig). Hence, we can guess that the Aβ cleaved from neurons signals to the astrocytes [23,30,31] to regulate cholesterol biosynthesis. Many studies have focused on cholesterol or cholesterol esters as risk factors for AD, and researchers are interested in how cholesterol levels affect AD by regulating Aβ production [29,37–39]. Taken together, both directions suggest a mutual regulation

mechanism between cholesterol biosynthesis and Aβ production (Fig 6E) that may play an essential role in normal biological function, such as synapse plasticity [40–45]. However, excessive production of Aβ causes toxic oligomers, plaques and activates brain inflammatory systems [46,47]. Additionally, cholesterol esters from excessive cholesterol in cells cause tau phosphorylation [48]. However, this may be an interesting hypothetical perspective and should be checked further by functional studies. Similar ideas have been suggested as the reciprocal modulations between Aβ production and cholesterol homeostasis in the cellular trafficking of cholesterol [49,50] and between APP and synaptic membrane cholesterol [51]. However, cholesterol biosynthesis was not central to their suggestions.

Would it make sense to simulate disease progress through gene expression changes by generating fake gene expression profiles and interpolating latent variables? This process is similar to the application of GANs to manipulate images by progressive conversion from male to female faces in GAN studies. We found that the transition curves for causal or similarly named genes showed correlated patterns. The underlying reason for this observation is thought to be that the gene expression values of 2M and 4M as well as 7M mice show similar patterns in the original rescaled RLD data. In other words, many collective data are assumed to participate in simulating the transition curves, which implies that accuracy of the complete data is essential in determining the transition curve patterns. Although it depends considerably on the quality of the original data, our study shows that simulating virtual disease progression and analyzing pathways through patterns can provide convincing evidence of the biological changes associated with AD progress.

Pathways such as cholesterol biosynthesis, the TCA cycle, and long-term potentiation, which usually involve a small number of genes, are dominant in one pattern because most of the transition curves of genes in these pathways exhibit similar trends. However, general pathways involving many genes consist of different patterns and are somewhat dominant in one or two patterns. Nevertheless, whether the different curve patterns of genes in a pathway imply biologically causal cascades requires further validation through functional approaches for each gene.

We simulated disease progression by latent space interpolation between two states in simple bulk RNA-seq, which is a similar idea as creating descriptions of cell differentiation using pseudo-time in scRNA-seq analysis. Considering the data sparsity problem and the laborious experimental preparation involved in scRNA-seq, the application of GANs to bulk RNA-seq is expected to offer the possibility of applying pseudo-time concepts more easily. Here, we suggest GANs may be a practical approach for predicting disease progression.

Indeed, it was easy to apply because the number of DEGs (1,208) was so suitable for our employed GAN architecture. With too many genes, it is difficult to obtain fake gene expressions with high similarity to the original data, and with too few genes, the information obtainable from a pathway analysis is poor. Hence, the method still requires further methodological improvements that will yield highly accurate fake data from larger numbers of genes. In order to utilize more genes, definitely, more high-quality data with systematic data augmentation in addition to GANs methodological improvements are necessary.

This work is based on the AD model mouse cortex data for 2M, 4M and 7M mice, which are publicly available. However, 5xFAD mice, an AD model adapted to the Aβ cascade hypothesis, do not fully recapitulate human AD, which is considered to be a multifactorial disease. In other words, the virtual disease progress used in this study can imply or describe early disease changes due only to Aβ overproduction. Due to this limitation when using animal model studies, the idea of using multiple datasets, which was used in the original scRNA-seq GANs work, might be more useful. Extending this GAN application of bulk RNA-seq to integrate multiple datasets is expected to require more careful normalization or methodology. As an interesting

example, if we were to combine RNA-seq data for the Aβ model mouse (5xFAD) data with the tau model mouse (PS19-P301S), the latent space interpolation might allow us to create a virtual mouse model with both properties—or to distinguish gene expression cascades between the two models. We hope that applying GANs to RNA-seq will be widely useful in disease research in terms of gene expression.

## Materials and methods

### Brief explanations of GANs

GANs are a machine learning algorithm that generate fake data based on given real data. They consist of two models, a generator and a discriminator, which generally use platforms of neural networks. The generator trains its network to generate samples similar to real data. The discriminator tries to distinguish between the generated and the real data as accurately as possible. Both networks perform adversarial learning to optimize their goals based on their loss function. Hence training GANs is a minimax optimization problem. Recently they have been highly paid attention among various generative models due to their continuous performance improvements and powerful applicability via vector arithmetic in latent space. Moreover the generator is thought to capture the semantic features of real data, which enables to generate real-like fake data with a new style that did not exist.

In this study, we try to generate gene expression profiles based on the real profiles for RAN-seq data of 5xFAD mice. The highly resembled fake data are essential to simulate interconversion between two states corresponding to Alzheimer's disease progression. The detailed network structure, hyperparameters, optimization and analysis procedures of our approaches are given in below sections (Network architecture and hyperparameters, Latent interpolation) and the GitHub repository (https://github.com/KBRI-Neuroinformatics/WGAN-for-RNASeq-analysis).

### Data augmentation

The real data set consists of a total 36 samples from six groups and each group consists of six samples. The data augmentation procedure was initiated based on the pairwise combinations within six samples belonging to a group. 15 pairwise combinations can be considered within six samples. Under a combination between two original samples (S1 and S2) (Fig 1), we created nine augmented samples ($S_{aug}$) by linear interpolation ($S_{aug} = xS1+(1-x)S2$, where $x = 0.1, 0.2, \ldots, 0.9$). Hence a total of 810 augmented samples (6 groups × 15 pairs × 9 linear interpolations) could be produced. Now we can use 846 samples (36 real + 810 augmented samples) for GANs training.

### Normalization for deep learning input scales

We applied the standard scaling operations for a sample(i), a gene(j) and a condition(k) as follows:

$$SS[i, j] = (RLD[i, j] - \mu[j])/\max(\sigma[k, j])$$

where $\mu[j]$ is the average of the j genes over all samples and $\sigma[k,j]$ is the standard deviation of j genes over the samples with a k condition. The reason using a maximum of $\sigma[k,j]$ is that we considered variations of gene expression within the same condition and excluded the effects of very tiny deviations within the same condition caused by the substantial differences between different phenotypes. The standard scaled RLD values were rescaled as rescaled RLD = SS/$(3.918\sigma_{SS})$+0.5 so that 95% of the values fell within the range [0,1], which is appropriate for the leaky ReLU activation function used in GANs.

Ghahramani *et al.*[9] used the min-max scaler to force all the real data values to the range [0, 1], which causes an innate mismatch with the generated data. The leaky ReLU activation function in the generator network continually yields values below 0 or above 1, which is a crucial hindrance in generating a highly correlated gene profile.

## Network architecture and hyperparameters

Because we reduced the number of target genes to 1,208 (7M DEG), the layout of the discriminator input units and generator output units needed to be modified. Through numerous trials and errors, we set the number of hidden layer units for the generator and the discriminator to 250 and 150, respectively, while the number of latent space units remained the same as in Ghahramani's work. Hence, the fully connected generator and discriminator network architectures were configured to be 100-250-250-1,208 and 1,208-150-150-1, respectively.

To optimize the network, we initialized the weight parameters in the generator randomly within the range [-0.3, 0.3]. We modified the process for generating random variables in the latent space: instead of using a combination of the Gaussian and the Poisson distributions, we generated random variables governed by the distribution of the rescaled RLD data. Those minor modifications were noticeably helpful for enhancing the training performance of this network.

## Latent interpolation

We randomly generated 10,000 latent vectors (z) at each epoch and gene expression profile vectors x = G(z) using the generator G. We obtained the top 10 high-correlation profile vectors $G(z)_{high}$ for an augmented sample, estimated the average latent space vector <z> over 10 G $(z)_{high}$, and generated an averaged profile G(<z>). In this way, we generated 846 resembled fake gene profiles G(<z>) corresponding to the 846 augmented data in each epoch—141 resembled fakes for each condition. A difference vector was calculated by subtracting the averaged latent vectors for each condition as $\Delta = \sum_i^{141} z_{AD7M(i)}/141 - \sum_i^{141} z_{WT7M(i)}/141$ for two 7M states. The simulated transitional states between WT and AD were estimated by the arithmetic equation $z(t,i) = z_{WT7M(i)} + t\Delta$ and the generator G(z(t,i)), where t = [0,1] and $z_{WT7M(i)}$ represents 141 latent vectors for 7M WT. We calculated the transition curves T(t) for 1,208 genes by averaging, where $T(t) = \sum_i^{141} G(z(t,i))/141$. The above averaging processes using all the augmented WT states were necessary due to the high sensitivity of the difference vectors ($\Delta$) and the irregular curve patterns, which sometimes occurred at the starting points ($z_{WT}$). Finally, we averaged the transition curves <T(t)> over 100 epoch points (75k~125k).

## Robustness of GANs training and analysis procedure

We checked whether our results were robust to changes of data augmentation method and variation of hyperparameters. In this study, we tried a wide variety of hyperparameters to get optimized performance of fake data. As some examples, results with different hyperparameters are shown in S10A and S10B Fig. It could be seen that GANs training processes were successful and the results were slightly different but overall similar. As mentioned in the above data augmentation section, we devised the data augmentation by linear interpolation, which is simple and easy to apply. However, there may be artifacts, and a different data augmentation method, which uses fitting original real data to gaussian distribution and random generation of augmented data by the distributions, was devised to verify robustness of the procedures. The results are shown in S10C Fig, which shows the training process is working well but the performance is slightly low. Hence we verified robustness of our results to variations of hyperparameters and change of data augmentation method.

## Supporting information

**S1 Fig. Temporal curves for network losses and average correlation of test data.** (**A**) The generator loss (red) and the discriminator loss (blue) throughout the 200k epochs. Training convergence begins after roughly 25k epochs. (**B**) The average pairwise Pearson correlation between 84 generated data and 84 test data matched by correlation values. The average maximum correlation reached approximately 80k.
(TIF)

**S2 Fig. Clustering patterns of tSNE plots.** (A) The tSNE plot made by the only original 36 samples shows the scattered sample points without clustering. (B) The tSNE plot made by the only augmented 810 samples shows the clear six groups representing different phenotypes. (C) The tSNE plots of the combined data (the original(36), the augmented(810) and the resembled fakes(36)). The original and the augmented 846 samples are visualized showing excellent clustering of the same groups. (D) The tSNE plot of the combined data including the resembled fakes (black stars) shows well overlaps between the original samples and resembled fakes.
(TIF)

**S3 Fig. The normalized Apoe gene expression values of real and fake data.** The rescaled RLD of the Apoe gene for the six 7M AD samples (blue and turquoise bars) and the resembled fakes corresponding to the third real data (turquoise bar) during the 75k to 125k epochs. The rescaled RLD values of the Apoe gene show continual variations over the epochs. The standard deviation of the temporal variations of the resemble fakes value is approximately 0.1, which is slightly larger than expected, although the correlation values can range as high as 0.95 ~ 0.99. At first glance, the deviations from the real values appear to be a random process of time. We checked the deviations based on the gene correlation heatmaps (**S4 Fig**).
(TIF)

**S4 Fig. The sample-specific and gene-specific heatmaps of genes. (Top)** The correlation heatmaps used to check the collective variations between genes were measured based on the temporal deviations of the 1,208 genes. The images show the correlation heatmaps of the genes obtained by temporal deviations over 7.5k to 125k epochs for a sample in 7M AD (left), the averaged temporal deviations of 6 samples in the 7M AD group (middle) and all 36 samples (right). The heatmap (left) for a sample shows more obvious collective features than do the averaged deviations over a group (middle) or all (right). In this way, we observed genes varying in the same pattern over the epochs, and they represent the properties of a sample rather than the properties between the genes. This indicates that the collective behavior of gene expressions were supposed to be learned and optimized to recognize sample-specific features rather than gene-specific features. **(Bottom)** The weight parameters of the last layer in the generator are supposed to contain gene-specific features as suggested by Ghahramani *et al.* in the scRNA-seq study. They suggested this GAN-derived gene association network is corresponding to non-linearly combined co-regulated genes and is distinct from linear and directly correlated regulations of a gene expression profile. Therefore, the correlation heatmap constructed by the weight parameters is predicted to show biological gene associations. The gene-specific heatmap for the 1,208 genes (Fig 2E) appears to be less collective than does the sample-specific heatmap (Top left). However, the collective gene associations in the sample-specific heatmap seems to be too specifically adapted for each sample and do not display appreciably as general gene association networks, while the gene-specific heatmap provides general features that are consistent with known coexpression networks. By using a cutoff of 0.4 in the heatmap, we obtained gene network clusters (left) and present the second-largest cluster to show the core upregulated genes related to the microglia immune system that were upregulated in our DEG

analysis.
(TIF)

**S5 Fig. The transition curves with the original data points for all months of the selected genes in Fig 3A and 3B.** The vertical curve widths represent the standard deviations of the latent interpolations over the 75k to 125k epochs. All the starting and ending points of the curves are in the middle of the original data points, indicating that the latent space interpolation works quite well. The plots show little change in 2M, which indicates that there is little difference between the AD and WT. However, we can observe differences in 4M and 7M. Some genes, such as Apoe and Abca1, have been shown to exhibit a delayed increasing pattern with little change during the first half and a sharp increase during the last half.
(TIF)

**S6 Fig. Transition curves and heatmaps of each gene for the selected pathways.** The microglial pathogen phagocytosis pathway shows that most genes follow a gradual increase in P2. The TCA cycle, which is significant in P5, and long-term potentiation, which is significant in P6, have a small number of genes showing homogeneous patterns. Other pathways related to the neurotransmitter transport and vesicle cycle show that the majority of genes follow a gradual decrease (P5) with lower FDR values in P5. However, several genes show early decreases in their transition curves. The pathways related to cognition and cholinergic synapse that many genes follow late decreases (P6) have lower FDR values in P6.
(TIF)

**S7 Fig. WikiPathway for cholesterol biosynthesis and fold changes of six genes.** (**Top**) All 15 genes in cholesterol biosynthesis are colored with log2FC values of AD7M and WT7M. The Sc4mol gene is the same as Msmo1. (**Bottom**) Plots of the fold changes of six genes that are in the 1,208 DEGs for 7M, in cholesterol biosynthesis. The normalized counts by DESeq2 for gene expression levels are used to evaluate the fold changes against a control value of the mean 7M WT. P-values $< 0.1$ (#), $0.01$(*), $0.001$(**) and $0.0001$ (***).
(TIF)

**S8 Fig. Scatter plots for six genes vs APP from human fusiform gyrus data.** Scatter plots of gene expression levels, which are normalized counts by DESeq2, for six human genes vs human APP. Positive correlations were observed for least four genes with $R > 0.4$ (MSMO1, HMGCS1, SQLE, and NSDHL). We performed RNA-seq analysis following the pipeline (Trimmomatic-HISAT2-HTSeq2-DESeq2) with GRCh38 annotations.
(TIF)

**S9 Fig. Cell-type gene expressions of six genes for mouse cerebral cortex.** The normalized gene expression levels (TPM) for the six genes in the eight cell-types of mouse cerebral cortex (GSE52564). We performed RNA-seq analysis following the pipeline (Trimmomatic-HISAT2-Stringtie) with GRCm38 annotations. OPC(oligodendrocyte precursor cells), NFO (newly formed oligodendrocytes), MO(myelinating oligodendrocytes).
(TIF)

**S10 Fig. Five measures for robustness checking in variation of GANs trainings.** Results of (A) linear data augmentation and hyperparameters (the numbers of hidden layer units of generator (HLUG = 200) and discriminator (HLUD = 120)), (B) linear data augmentation and hyperparameters (HLUG = 300 and HLUD = 18), and (C) gaussian data augmentation and hyperparameters (HLUG = 250. And HLUD = 200). Results are shown by five measures as below. The average pairwise Pearson correlation between 84 generated data and 84 test data corresponding to S1 Fig. The distribution plots of all rescaled RLD values for the 846

augmented real samples (blue) and the 846 generated samples (red) corresponding Fig 2A. Correlation coefficient distributions for all pairs within the 846 real data (blue) and for pairs between the 846 real and 846 fake data (red) at the 100k epoch corresponding Fig 2B. tSNE plots at four epochs with different colored dots representing the 762 training (red) samples, 84 test (blue) samples and 84 generated samples (orange) corresponding Fig 2C. Transition curves of selected four genes from 7M WT to 7M AD corresponding Fig 3A and 3B.
(TIF)

**S1 Table. DEGs for 2M, 4M and 7M.** Three sheets by month include scaled counts per pheno-type, log2FC, p-value and adjusted p-value, which are evaluated by DESeq2 package.
(XLSX)

**S2 Table. WikiPathway, KEGG, and GOBP results by WebGestalt.** Eight sheets by patterns (Up, Down, P1, P2, P3, P4, P5, P6) include significant pathways (FDR <0.05) with enrichment ratio, p-value, FDR and genes. Two additional sheets show 50 selected pathways to compare results by patterns.
(XLSX)

## Author Contributions

**Conceptualization:** Mookyung Cheon.

**Formal analysis:** Jinhee Park, Hyerin Kim.

**Investigation:** Jaekwang Kim, Mookyung Cheon.

**Methodology:** Jinhee Park, Mookyung Cheon.

**Software:** Jinhee Park.

**Writing – original draft:** Jaekwang Kim, Mookyung Cheon.

**Writing – review & editing:** Mookyung Cheon.

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
