## [Decision Letter · Decision Letter 0]

29 Feb 2020

Dear Dr. Cheon,

Thank you very much for submitting your manuscript "A practical application of generative adversarial networks for RNA-seq analysis to predict the molecular progress of Alzheimer's disease" for consideration at PLOS Computational Biology.

As with all papers reviewed by the journal, your manuscript was reviewed by members of the editorial board and by several independent reviewers. In light of the reviews (below this email), we would like to invite the resubmission of a significantly-revised version that takes into account the reviewers' comments.

Your revision must address all the issues raised by the reviewers and including the points related to reproducibility, robustness and software and data file documentation.  Moreover, please be sure to improve the presentation of GANs to a non-expert audience, and to justify their application for gene expression classification.

We cannot make any decision about publication until we have seen the revised manuscript and your response to the reviewers' comments. Your revised manuscript is also likely to be sent to reviewers for further evaluation.

Sincerely,

Hugues Berry

Associate Editor

PLOS Computational Biology

William Noble

Deputy Editor

PLOS Computational Biology

Reviewer's Responses to Questions

**Comments to the Authors:**

Reviewer #1: This paper presents an application of generative adversarial networks (GANs) to predict the molecular progress of Alzheimer’s disease. It is a topic of interest to researchers in this related field, but the paper needs very significant improvement.

1. In general, there is a lack of explanation of the method of GANs used in this study. The author should provide more detailed information about GANs to make the model clearer for the reader.

2. The results are a little hard to follow for someone who is not an expert in this field or without closely reading the methods.

3. Are the results robust? For example, whether different data augmentation method or parameters can lead to large changes in results.

Reviewer #2: Summary:

Authors present in silico study of the molecular mechanism of Alzheimer's disease progression from wild type to disease state using publically available mouse model RNAseq data.

They raise the problem of lacking sufficient experimentally generated data to study various stages of the disease progression and propose to address it by producing synthetic data using generative adversarial networks (GANs) that have become popular deep learning methods in the fields such as image analysis and being adapted for biological studies. In the manuscript authors firstly demonstrate data generation process, then they analyse introduced disease transition curves in the context of disease related pathways. Authors perform pathway enrichment analysis using interpolated data of up- and down-regulated genes and suggest the hypothesis of mutual regulation between amyloid beta production and cholesterol biosynthesis. The manuscript is supported by the analysis code provided via GitHub repository https://github.com/KBRI-Neuroinformatics/GAN-for-bulk-RNAseq.

I am recommending the paper could be accepted after authors have addressed the issues stated in the major and minor comments sections. My major concern is the reproducibility of the results demonstrated in the article. Major and minor comments are provided below.

Major comments:

1. It is not currently mentioned if a power analysis and gene effect sizes were taken to account when identifying a significantly differentially expressed genes.

2. It is also not clear whether 1208 DEGs were differentially expressed only between 7MWT vs 7MAD or other comparison was used to identify these genes.

3. It is not quite clear what causality authors intend to discover on p.4 line 96 and how it is reflected in the results part.

4. P.5 line 112; p.19 line 430 Please clarify whether during the augmentation procedure the interpolation was performed between samples belonging to one group or from different groups. Figure 1 caption is a missing explanation of what s1...s2 is.

5. Please clarify the choice of data generated at 100k epoch for the comparison with the original augmented data instead of data generated at 25k epoch when convergence has already been reached.

6. Model performance evaluation metrics such as precision, recall and F1 score are not presented.

7. Please explain why only 846 samples were generated by GANs.

8. P.8 line186 - p.9 line193. It is quite challenging to evaluate the claims that authors make about the gene-specific features due to the construction of the sentences.Figure S3 that is referred to as S3A and S3B in fact does not have A and B part, legend and caption.

9. It’s also not clear how collective behavior in the gene association network is connected with the weight parameters of the last layer in the generator.

10. P.9 line 209. It is not clear why only transition curves from 7M WT to 7M AD were selected for further analysis. Also the caption of Figure 3(A-B) is misleading, i.e. it does not provide information that only transition curves from 7M WT to 7M AD were plotted. Please clarify what is meant by “original data points for all months” on Figure S4.

11. For transition curves classification authors use assumption of the existence of 6 predefined disease trajectories. It is not clear if this assumption is based on the previously confirmed domain knowledge or authors propose their hypothesis.

It’s not clear what is TYROBP causal network is and how it was identified.

Software:

1. The software is not sufficiently documented to fully reproduce the analysis. I encourage authors to add user guidelines and requirements (e.g. python 3, etc.) to the README file.

2. Instead of referring to the original dataset GSE104775 from the dedicated repository authors provide their own composed data file GSE104775_RLD_OurDEG_ADWT_bM100_augmentation_rev.npz that is not accompanied by a clarification of what is its content. I suggest authors to add a corresponding description of the data file deposited at github repository.

3. The file GAN_for_bulkRNAseq.ipynb does not provide differential expression analysis that is the first step of the analysis described in the manuscript. I encourage authors to add missing analyses scripts to fully reproduce the results starting from the downloading of the raw data, preprocessing and differential expression analysis.

4. Analyses of GSE90693 and GSE125583 data sets used for the validation of biological findings are missing from the project repository.

Minor comments:

1. In order to increase the reproducibility of the results I suggest the authors to share software on GitHub with specified release version and accompanied by a code DOI. It can be obtained at DOI providing repository such as Zenodo or similar. This would provide permanent access to a usable instance of the published code. Code with an assigned DOI may be formally cited in future publications.

2. Finally, I noticed the study source code at https://github.com/KBRI-Neuroinformatics/GAN-for-bulk-RNAseq was not available under an open source license. An open source license is essential to allow other researchers to modify and reuse the code. I recommend the authors release the software under a permissive open source license, such as the MIT License. See https://help.github.com/en/github/creating-cloning-and-archiving-repositories/licensing-a-repository.

3. I recommend authors to carefully read their text and pay attention to punctuation.

I found few typos cases where the references were placed after the end of the sentence, e.g. p.3 line 44, 53, 54, 56; p.4 line 78, 90,105, etc. A dot is missing the end of the sentence on p. 9 line 193. Typo on p.13 line 27.

4. I haven't quite understood the meaning of the sentence on p. 1 line 20-24 in the context of information given in the abstract.

Reviewer #3: The authors used generative adversarial networks (GANs) to capture the gene expression transition patterns from wild-type samples (WT) to Alzheimer’s disease (AD) samples. To train the deep neural net, the author did data augmentation and selected differentially expressed genes as the features. Evaluation methods, including Pearson correlations between real and fake samples, distribution of the fake samples compared to real samples, and a t-SNE clustering after training convergence, were used to demonstrate the validity of the augmented samples. The author used the latent space interpolation of the GANs and showed 6 patterns of AD progression, which provide new insights in dissecting biological (AD) pathways. In general, the manuscript is well organized and the study is rigorous, but we have a few concerns:

1. It would be more convincing by showing additional t-SNE clustering results for the original 36 samples and the augmented samples and mind: If either of the t-SNE results shows similar clusters to Figure 2C. If the answer to question (a) is a yes, for every cluster, how many of the genes overlap with those when use all samples including real, augmented and GAN generated samples?

2. By performing the analysis mentioned above, I would be either convinced or skeptical about the necessity of using GANs to find the patterns, though GANs can make the transition plots.

3. In the paper, the author put forward the current problem of limited availability of biological samples. But it’s not their deep learning model that solved this problem. What indeed deals with this problem is the process of data augmentation in their method. There might be a misleading claim in the paper that their deep learning model solved the limited sample size problem.

4. The author built a GANs model to simulate gene expression data of 846 real expression data. After showing their fake data simulates the real data pretty well, they tried to interpret the generative network. From their learned model, by using latent space interpolation method in GANs, they were able to classify 6 patterns of gene expression change from WT to AD state. Then they did downstream analysis like pathway analysis to learn the pathological progress of Alzheimer’s disease. But why GANs? Traidtional1. bioinformatics tools could also classify gene expression change patterns since they have time series data for both WT and AD mice. Here the author failed or didn’t show that GANs is better than bioinformatics method, or using GANs is a necessity.

**Have all data underlying the figures and results presented in the manuscript been provided?**

Reviewer #1: Yes

Reviewer #2: Yes

Reviewer #3: Yes

PLOS authors have the option to publish the peer review history of their article (what does this mean?). If published, this will include your full peer review and any attached files.

Reviewer #1: No

Reviewer #2: No

Reviewer #3: No
---

## [Decision Letter · Decision Letter 1]

28 Jun 2020

Dear Dr. Cheon,

We are pleased to inform you that your manuscript 'A practical application of generative adversarial networks for RNA-seq analysis to predict the molecular progress of Alzheimer's disease' has been provisionally accepted for publication in PLOS Computational Biology.

Best regards,

Hugues Berry

Associate Editor

PLOS Computational Biology

William Noble

Deputy Editor

PLOS Computational Biology

Reviewer's Responses to Questions

**Comments to the Authors:**

Reviewer #1: The issues have been addressed.

Reviewer #2: Manuscript:

Revised version of the manuscript has been substantially improved over previous submission.

In sections where I am qualified to evaluate I did not find any issues.

The provided approach is potentially of genuine utility.

Software:

Inclusion of the scripts for all steps of the analysis and user guidelines for the software clearly increase the value of the work for the research community and contributes to research reproducibility.

Reviewer #3: All concerns addressed.

**Have all data underlying the figures and results presented in the manuscript been provided?**

Reviewer #1: None

Reviewer #2: Yes

Reviewer #3: None

PLOS authors have the option to publish the peer review history of their article (what does this mean?). If published, this will include your full peer review and any attached files.

Reviewer #1: No

Reviewer #2: No

Reviewer #3: No

---

## [Editor Report · Acceptance letter]

16 Jul 2020

PCOMPBIOL-D-19-02173R1 

A practical application of generative adversarial networks for RNA-seq analysis to predict the molecular progress of Alzheimer's disease

Dear Dr Cheon,

I am pleased to inform you that your manuscript has been formally accepted for publication in PLOS Computational Biology. Your manuscript is now with our production department and you will be notified of the publication date in due course.

With kind regards,

Sarah Hammond
